# Efficacy and Safety of a Single Ivy Extract Versus Two Herbal Extract Combinations in Patients with Acute Bronchitis: A Multi-Center, Randomized, Open-Label Clinical Trial

**DOI:** 10.3390/ph18050754

**Published:** 2025-05-20

**Authors:** Peter Kardos, Justus de Zeeuw, Inga Trompetter, Simon Braun, Yuliya Ilieva

**Affiliations:** 1Lung Centre Maingau, 60316 Frankfurt am Main, Germany; 2Pneumologische Praxis, 51105 Köln, Germany; 3Engelhard Arzneimittel GmbH & Co. KG, 61138 Niederdorfelden, Germanys.braun@engelhard.de (S.B.)

**Keywords:** bronchitis, Ivy, cough, Thyme, Primrose, EA 575, Prospan^®^ Cough Drops, Bronchipret^®^ Drops, Bronchicum^®^ Drops

## Abstract

**Background**: The combination therapy for acute bronchitis with several plant extracts, such as Ivy and Thyme or Primrose and Thyme, is assumed to offer added benefit over single extract preparations. However, no clinical trials have yet demonstrated such a therapeutic advantage. **Methods**: In this three-arm, open-label, randomized clinical trial, patients with acute bronchitis were assigned to groups receiving Ivy extract EA 575 (Prospan^®^ Cough Drops), Ivy/Thyme extract combination (Bronchipret^®^ Drops), or Thyme/Primrose extract combination (Bronchicum^®^ Drops) according to their respective labels. The primary endpoint was the assessment of non-inferiority, and the second endpoint was the assessment of superiority of Ivy vs. each of the two comparators (Ivy/Thyme and Thyme/Primrose) regarding the change in Bronchitis Severity Score between baseline and day 7. In total, 325 adult patients were considered for evaluation. **Results**: Non-inferiority of Ivy extract was statistically significant against both comparators (both *p* < 0.0001). Superiority of Ivy extract was statistically significant against Ivy/Thyme extract (*p* < 0.0001) but missed statistical significance against Thyme/Primrose extract (*p* < 0.0607). The incidence of adverse events was low and comparable between the groups. All adverse events were non-serious. **Conclusions**: these data revealed that Ivy extract EA 575 is non-inferior in acute bronchitis treatment compared to both comparators and superior to Ivy/Thyme.

## 1. Introduction

The incidence of cough is high and, in Western primary care, it is estimated to be 12.5%, with respiratory tract infections and bronchitis accounting for the most frequent underlying conditions [1]. Up to 95% of cases are attributable to respiratory viruses, which also cause the “common cold”. This entity is an umbrella term for upper airway viral infections [2,3,4]. Although acute bronchitis can occur as an isolated condition, it is typically indistinguishable from the common cold in clinical practice [5]. 

While acute bronchitis is often regarded as a minor illness, it can still have a substantial impact on a patient’s health-related quality of life [6]. 

Although it is usually a self-limiting disease, and typically resolves within 1 to 3 weeks, patients may benefit from an effective symptomatic treatment. There are several treatment options to shorten the duration and alleviate the intensity of acute bronchitis. These drugs consist mainly of over-the-counter herbal remedies and synthetic mucoactive drugs, for which there is, however, a variable degree of clinical evidence regarding their efficacy and tolerability [7,8]. 

Combinations of multiple plant extracts, such as Ivy and Thyme or Primrose and Thyme, intuitively appear to provide therapeutic benefits compared to single-extract preparations. Particularly for cough treatment, a combination of Thyme (*Thymus vulgaris*) extract with either Ivy (*Hedera helix*) or Primrose (*Primula veris* root) extracts has been postulated to have an additive effect; although, to our knowledge, no clinical trials have yet demonstrated such a therapeutic advantage [9]. Thus, an open-label, randomized head-to-head study was designed comparing the efficacy of two combination preparations and one single extract preparation that are commonly used in clinical practice for the treatment of acute bronchitis and/or acute cough associated with the common cold. The study design had to consider that different herbal preparations containing extracts of the same plant may vary considerably in terms of efficacy and tolerability, due to the natural variability of the harvested plants, but also due to factors associated with the manufacturing process [8]. Ivy, Thyme, or Primrose extracts are multi-constituent plant extracts, containing several phytochemical classes with pharmacological significance in varying amounts. For Ivy, these include saponins, flavonoids, and dicaffeoylquinic acids [10]. For Thyme, phenolic compounds, terpenoids, and flavonoids have been described [11], while Primrose’s main components comprise saponins and phenolic compounds [12]. Therefore, results of a clinical trial comparing specific herbal preparations cannot be generalized to the effects of the plant in general. Instead, head-to-head comparisons of marketed products are necessary. 

The general efficacy of all three investigated extracts/extract combinations has previously been shown in placebo-controlled trials [13,14,15,16,17]. Accordingly, the prescription of these drugs has been recommended by the current cough guidelines [8]. To complement these data, the presented trial directly compared the efficacy and safety of three marketed herbal preparations in adults with acute bronchitis. First, the goal was to determine whether the single Ivy extract EA 575 is non-inferior in alleviating the patients’ disease burden compared to each of the two combinations (Ivy/Thyme and Thyme/Primrose). This would help confirm whether combining multiple herbal extracts necessarily offers additional benefits. In the event that non-inferiority can be demonstrated, subsequent superiority testing will be conducted to assess and explore the potential advantages of the Ivy extract EA 575 in the treatment of bronchitis compared to the respective herbal combination comparator.

## 2. Results

### 2.1. Study Population

In total, 330 patients were screened. Of these, 328 patients were randomized. The prospectively defined, unequal allocation between the study arms resulted in a group of n = 140 Ivy patients and 188 non-Ivy patients distributed in two groups (Ivy/Thyme: n = 93, Thyme/Primrose: n = 95). Three patients discontinued the study prematurely and were excluded from the modified Full Analysis Set (mFAS). The mFAS thus comprised 325 patients. Four patients were excluded from the PPS, including two patients from the Ivy group and one patient each in the Ivy/Thyme and Thyme/Primrose group. The PPS thus included 321 patients (Figure 1).

The randomized patients were equally distributed over the study sites and treatment groups. The treatment groups did not show any relevant differences in terms of demographics, baseline characteristics, relevant medical history, or concomitant medication (Table 1). All patients confirmed taking the IMP regularly as instructed at all treatment visits.

### 2.2. Efficacy Results

#### 2.2.1. Confirmatory Results

In the PPS, the mean BSS change from baseline to day 7 was −9.11 score points (SD: 2.44) in the Ivy group, −7.64 points (SD: 2.68) in the Ivy/Thyme group, and −8.59 points (SD: 2.45) in the Thyme/Primrose group. Thus, the non-inferiority of Ivy extract was shown to be statistically significant against both comparators: against Ivy/Thyme extract (LS means treatment effect for change: 1.48 points, 95% CI: −2.08; −0.89, *p* < 0.0001) and against Thyme/Primrose extract (LS means treatment effect for change: −0.56 points, 95% CI: 1.14; 0.02, *p* < 0.0001, Figure 2). Hence, both primary endpoints could be reached in the first, a priori, ordered steps.

Next, the superiority of Ivy extract was statistically significant against Ivy/Thyme extract in the mFAS (LS means treatment effect for change: −1.45 points, 95% CI: −2.04; −0.85, *p* < 0.0001). However, against Thyme/Primrose extract, the secondary endpoint of superiority missed the statistical significance (LS means treatment effect for change: 0.56 points, 95% CI: −1.14; 0.03, *p* < 0.0607) in the mFAS (Figure 2). Additional sensitivity analyses were conducted using the mFAS for the non-inferiority comparisons and the PPS for the superiority comparisons. The results did not differ essentially from those of the confirmatory analyses.

#### 2.2.2. Exploratory Analyses

Mean change in BSS from baseline to all post-baseline visits was more pronounced in the Ivy group compared to the Ivy/Thyme and Thyme/Primrose group, respectively, for the 7-day treatment phase and over the further 7-day post-treatment observational period (Figure 3a). This pattern of results is also reflected in the overall symptom burden over the entire treatment and observational period, indicating the lowest overall symptom burden in the Ivy group (*p* < 0.05 for all post-baseline visits) (Figure 3b).

The threshold of a MCID (3.15 points on the BSS) was surpassed on day 2 for the Ivy group (mean change: −3.66 points) and on day 3 for the Ivy/Thyme (mean change: 3.95 points) and Thyme/Primrose (mean change: −4.41 points) groups (Figure 4a). 

In a post hoc analysis, the proportion of patients was determined for whom the presence of bronchitis was considered unlikely (recovery, BSS < 3, [18], Figure 4b). Accordingly, on day 7, 55.4% of patients in the Ivy group, 34.1% of patients in the Ivy/Thyme group, and 48.4% of patients in the Thyme/Primrose group were considered as “recovered”.

The superiority of Ivy extract over both Ivy/Thyme extract and Thyme/Primrose extract, respectively, was furthermore demonstrated for cough severity (VAS) for all post-baseline visits (Figure 5). 

The results of cough severity (VAS) on days 7 and 14 are in line with cough severity assessments by VCD categories (Table 2).

For both patients and investigators, global efficacy was judged better for Ivy extract compared to Ivy/Thyme extract and Thyme/Primrose extract, respectively, on days 7, 10, and 14 (all *p* < 0.0001) (Figure 6 and Appendix A).

### 2.3. Safety Results

Thirteen adverse events were reported in nine patients. Three adverse events were reported in the Ivy group (2.1%), one adverse event in the Ivy/Thyme group (1.1%), and nine adverse events in the Thyme/Primrose group (9.5%). A possible relationship was documented in two cases: “diarrhea” (Ivy group) and “dyshidrotic eczema” (Thyme/Primrose group). Furthermore, the AE “throat irritation” (Thyme/Primrose group) was assessed as certainly drug-related by the investigator. There were no deaths or other serious adverse events during the study. 

The global tolerability of the study drug was assessed similarly by patients and investigators on days 7 and 14 (Appendix A).

## 3. Discussion

The trial aimed to assess whether the single extract EA 575 is non-inferior or potentially superior to each of the two combination treatments (Ivy/Thyme and Thyme/Primrose) in alleviating bronchitis symptoms. 

Ivy extract EA 575 was non-inferior to both dual-component comparators (Ivy/Thyme extract, Thyme/Primrose extract) based on the change of the BSS after seven days of treatment. The primary study objective was therefore fully met.

On treatment day 7, Ivy extract showed a significantly greater BSS-reduction than Ivy/Thyme, and a greater, but not statistically significant, reduction than Thyme/Primrose extract. Thus, the confirmatory secondary endpoint of superiority was also partially met. 

Additionally, the superiority of Ivy extract is reflected by the exploratory analysis of the overall symptom burden: the cumulative BSS over the entire treatment and observational period showed the lowest overall symptom burden in the Ivy group (statistically significant at all treatment visits). Similarly, from day 2 onwards, the cough severity (VAS) was statistically lower at all visits in the Ivy group compared to Ivy/Thyme and Thyme/Primrose. 

The clinical relevance was determined by applying the generic recommendation for a minimal clinically important difference (MCID) endorsed by the IQWiG (i.e., a change of 15% of the total range of a given scale) [19]. For the BSS change, this translates to a mean improvement of 3.15 points, which was surpassed already from day 2 on in the Ivy group, when the reduction in the BSS for Ivy/Thyme and Thyme/Primrose groups was not yet clinically relevant [20]. These findings are consistent with the performed post hoc analysis (evaluating the number of patients showing recovery, defined as a total BSS score of < 3 points, corresponding to a clinical interpretation of acute bronchitis being unlikely), which suggests an earlier onset of clinical recovery with Ivy treatment compared to the Ivy/Thyme or Thyme/Primrose treatment.

The results of the bronchitis symptoms determined using the BSS are, moreover, in agreement with the global efficacy assessment, both from the perspective of patients and physicians, who rated the treatment with single Ivy extract superior to Ivy/Thyme extract and Thyme/Primrose extract treatment, respectively.

The results of this study are also largely supported by the literature: in previous clinical trials comparing Ivy and Ivy/Thyme extracts against placebo, there was a significant difference from placebo from day 2 onwards for Ivy extract [13], and from day 4 onwards for Ivy/Thyme [16]. Even though this head-to-head trial did not include a placebo arm, these data are consistent with our analysis of clinical relevance (BSS reduction of ≥3.15): a clinically relevant improvement in our study occurred from day 2 onwards in patients treated with Ivy, and from day 3 in patients treated with Ivy/Thyme.

In a double-blind trial on the efficacy and tolerability of Thyme/Primrose [17], a reduction in the BSS from 12 to 6 was shown at day 4, i.e., data corresponding closely to those in our study (mean BSS at baseline: 11.79, mean BSS at day 4: 5.53) appears to indicate that the open-label design did not result in a significant distortion of the data due to potential bias. Notably, the same trial also assessed the onset of the treatment effect of thyme/primrose (based on the patients’ subjective assessment). The onset of treatment effect was determined to occur at 3.4 days in the verum group (versus 5.6 days in the placebo group) [17]. Again, these results are consistent with the data presented here, which showed a clinically relevant improvement starting from day 3 in the Thyme/Primrose group.

Previous preclinical studies have shown that components of the Ivy extract EA 575 prevent the internalization of β2 adrenergic receptors under stimulating conditions [21,22]. This results in an increase in intracellular cAMP levels, leading to bronchodilation and surfactant-mediated secretolysis as downstream positive effects [23]. In cell culture experiments, it has already been shown that a fixed combination of Ivy and Thyme (corresponding to the herbal preparation of Bronchipret^®^) unexpectedly does not result in additive or synergistic activation of β2-signaling [23]. In contrast to the cells pretreated with Ivy extract, pretreatment with Thyme extract or a Thyme–Ivy extract combination did not result in increased cAMP levels, and the number of β2 receptors on the cell surface was significantly reduced. The authors concluded that components of the thyme extract may at least partially counteract the positive effects of the ivy extract. The evidence of possible antagonistic interactions between the various compounds of plant extracts is in line with our data, demonstrating clinical benefits with single extract treatment. 

To our knowledge, this is the first study to compare head-to-head single Ivy therapy versus combination therapies. While the effectiveness of all three therapies has been demonstrated in previous research, the main goal of this study was not to prove their efficacy against placebo [13,14,15,16,17]. Instead, it aimed to compare the effectiveness of a single ivy extract with that of herbal combination products. The overall patient cohort covered a large demographic range of patients aged 18–75 years. Baseline data according to age, gender, and initial BSS of the three groups were well balanced, suggesting an effective randomization procedure. 

In this study, particularly frequent observations (i.e., daily visits from day 0 to day 4 and subsequent visits at days 7, 10, and 14) were carried out to ensure comprehensive monitoring in the short time period of this acute disease in order to increase the validity of the results. 

Due to the very different dosing regimens and compositional characteristics, effective blinding would have required a triple-dummy design, which was considered unduly burdensome and prone to self-medication errors for patients. Various types of potential biases towards the different IMPs were considered (e.g., detection bias, performance bias). A personal preference for one of these well-known OTC products among individual patients cannot be entirely excluded. However, given the fully randomized allocation of patients to the study arms and the sufficiently large comparison groups, any such preferences are likely to have been evenly distributed across all three groups, minimizing the likelihood of selective influence on the final results. The absence of major differences in outcomes between the different sites also suggests effective control of bias and enhances the generalizability of the data. 

In sum, despite the inevitable potential for bias on account of the open-label study design, it is presumed that the risks are minimal, and the validity of the study’s findings is provided.

## 4. Materials and Methods

### 4.1. Study Design

This was a randomized, controlled, open-label, multi-center, interventional clinical trial with three treatment groups (Prospan^®^ Cough Drops (Ivy, Prospan^®^ Hustentropfen containing ivy extract EA 575), Bronchipret^®^ Drops (Ivy/Thyme, Bronchipret^®^ Tropfen), and Bronchicum^®^ Drops (Thyme/Primrose, Bronchicum^®^ Tropfen)). The trial was carried out between February and April 2024 at ten active medical sites in Germany. The study was conducted in accordance with the principles of good clinical practice, the Declaration of Helsinki and applicable EU and German laws and regulations, following approval of the study protocol by the German Federal Institute for Drugs and Medical Devices (BfArM) and of the competent Ethics Committee of the Medical Association of the German federal state Hessen (Ethik-Kommission bei der Landesärztekammer Hessen) for all participating sites. The trial was registered and published under the identifier number EudraCT 2023-507370-41-00 in the official Clinical Trial Information System (CTIS) of the European Medicines Agency (EMA).

### 4.2. Patients

Key inclusion criteria were diagnosis of acute bronchitis with symptoms present for 48–72 h prior to study inclusion; male or female patients of any ethnic origin aged 18–75 years; a cough severity score of ≥50 mm on the 100 mm Visual Analog Scale (VAS), a Bronchitis Severity Score (BSS) of ≥10 points, and a Verbal Category Descriptive (VCD) score of ≥2 points at study inclusion. 

The key exclusion criteria were known bronchial hyper-responsiveness, chronic bronchitis, other chronic or hereditary lung diseases, asthma, severe allergic diseases, history of gastrointestinal bleeding, significant cardiovascular, hepatic, or renal disease, and a history of chronic gastritis or peptic ulcer. Excluded co-medications were corticoids, beta-2 agonists, expectorants, theophylline, antitussives, anesthetics, acetylsalicylic acid dose over 100 mg daily or other non-steroidal anti-inflammatory drugs, leukotriene inhibitors, ACE inhibitors, antiviral drugs or antibiotics, antihistamines, immunosuppressants, isoprenaline, atropine, sodium cromoglycate 7 days prior to inclusion in the study; any other phytopharmaceutical medication or homeopathic medicines for the common cold 7 days prior to inclusion in the study. Use of any of the above-mentioned medications led to exclusion throughout the entire study period. 

### 4.3. Study Medication and Randomization

The study drugs were administered according to the respective summary of product characteristics of the study medication: Prospan^®^ Cough Drops (Ivy), 24 drops orally 3 times a day, resulting in a daily dose of 50.4 mg of Ivy leaf dry extract; Bronchipret^®^ Drops (Ivy/Thyme) extract 2.6 mL orally 3 times a day, resulting in a daily dose of 3.9 mL Thyme herb liquid extract and 0.234 mL Ivy leaf extract or Bronchicum^®^ Drops (Thyme/Primrose) 35 drops orally 5 times a day, resulting in a daily dose of 2.2 g thyme herb liquid extract and 1.1 g primrose root tincture, each for a duration of 7 days.

Eligible patients were randomized at the baseline visit to one of the three treatment groups at a 3:2:2 ratio (Ivy: Ivy/Thyme: Thyme/Primrose) according to a list of unique random numbers. The IMP was consecutively numbered according to the randomization schedule, and the patient received the lowest available randomization number at the site, printed on the bottle label. To avoid a potential bias introduced by different product presentations via the packaging, the study medication was delivered in neutrally labelled packaging. To further minimize potential bias, investigators and site staff were trained on professional and impartial communication with study participants, aiming to prevent any influence on patients’ assessments. 

### 4.4. Evaluation Schedule and Study Endpoints

The study comprised a 7-day treatment phase followed by a 7-day observation phase. Study visits took place at baseline (day 0), during the treatment phase on days 1 to 4 and day 7, as well as during the observation phase on days 10 and 14 (end of study).

The primary endpoint was the assessment of non-inferiority of Ivy extract vs. each of the two comparators (Ivy/Thyme extract and Thyme/Primrose extract) regarding the change in BSS between baseline and day 7 (end of treatment phase). The BSS is a validated tool to measure the severity of acute bronchitis [18] and comprises the five key symptoms of acute bronchitis (coughing, sputum, rattling noise (crackles), thoracic pain whilst coughing, and dyspnea). The total sum score ranges between 0 and 20 points. The major secondary endpoint (confirmatory) was the assessment of the superiority of Ivy extract vs. each of the two comparators regarding the change in BSS between baseline and day 7.

The primary and secondary hypotheses of non-inferiority and superiority, respectively, were tested using the concept of hierarchically ordered hypotheses in an a priori defined order: e.g., superiority of Ivy extract vs. Ivy/Thyme extract was tested only if non-inferiority of Ivy extract vs. each of the two comparators had been shown.

Furthermore, the following secondary efficacy endpoints (exploratory) were defined:Change in BSS between baseline and all other post-baseline visits,The cumulative BSS between baseline and all post-baseline visits,Change in cough severity VAS score between baseline and all post-baseline visitsCough severity VCD score at all visits,Global efficacy assessment by patient using a five-point rating scale (0, very well; 1, well; 2, fair; 3, poor; 4, very poor) on days 7, 10, and 14.

Descriptive post hoc analyses furthermore included quantification of BSS recovery, defined as a BSS total score of <3 points on day 7, corresponding to a clinical interpretation of acute bronchitis being unlikely [19]. 

Safety endpoints included the incidence and type of adverse events, vital signs (systolic and diastolic blood pressure, pulse rate, and body temperature), and global tolerability assessed by the patient and investigator.

Medical history and adverse events were coded according to MedDRA (version [27.0]) and concomitant medications by ATC. 

### 4.5. Statistical Analysis

A fifteen percent change in the range of a scale can be regarded as a minimal clinically important difference (MCID), which has been proposed by the German Institute for Quality and Efficiency in Healthcare (IQWiG) [20]. In the context of the BSS, 15% equates to 3.15 points on the BSS. 

Half of 3.15 (1.5) was chosen as an a priori for the non-inferiority margin in this study. Furthermore, for sample size calculation, the corresponding standard deviation of the change was estimated to be three score points. For a two-sided non-inferiority test (α = 5%) with a statistical power of 90% and a treatment group allocation ratio of 3:2:2, a number of 249 patients was calculated as needed. To account for possible dropouts, the number of patients to be recruited was set at 300.

Non-inferiority of Ivy extract was tested separately against Ivy/Thyme extract and Thyme/Primrose extract. If non-inferiority could be proven, the superiority of Ivy extract was further tested against Ivy/Thyme extract and Thyme/Primrose extract, respectively. These statistical tests were performed in a confirmatory sense by means of an analysis of covariance (ANCOVA) with the factor treatment, covariate baseline BSS, and cofactor center with a significance level of α = 5%. Non-inferiority analyses were carried out for the per-protocol set (PPS) evaluation of superiority for the modified full analysis set (mFAS). All other statistical tests (ANCOVA for quantitative variables and Wilcoxon test for ordinal variables) were performed two-sided with a significance level of α = 5% on an exploratory basis for the mFAS.

The mFAS included all randomized patients who fulfilled all major entry criteria with an impact on validity of the primary endpoint, who received the IMP at least once, and provided at least one data value necessary post-baseline for the primary endpoint. 

The PPS included all randomized patients who completed the clinical trial without major protocol deviations.

All statistical analyses were conducted using the SAS statistical package for Windows 10 (version 9.4, SAS Institute, Cary, NC, USA).

## 5. Conclusions

While herbal multi-extract combinations are often credited with efficacy advantages over single-extract preparations, this study provides clinical evidence that combinations of different extracts do not necessarily offer therapeutic benefits. Instead, the study confirmed the premise that the efficacy of each plant extract and each extract combination must be individually demonstrated. 

Specifically, this study shows that the efficacy of Ivy extract EA 575 in the treatment of acute bronchitis is non-inferior to that of Thyme/Primrose and superior to that of Ivy/Thyme.

## Figures and Tables

**Figure 1 pharmaceuticals-18-00754-f001:**
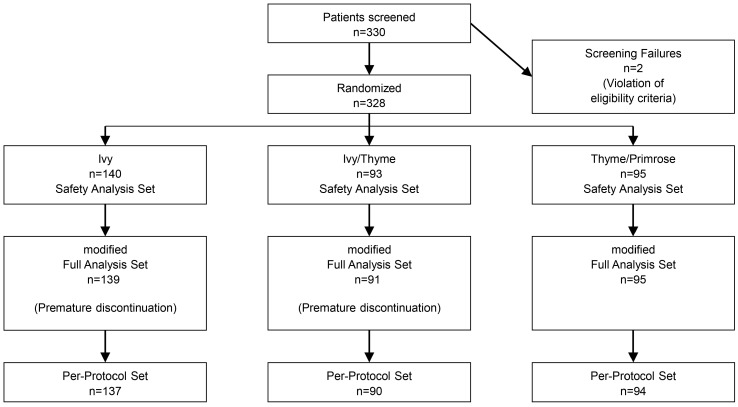
Disposition of patients and analysis sets.

**Figure 2 pharmaceuticals-18-00754-f002:**
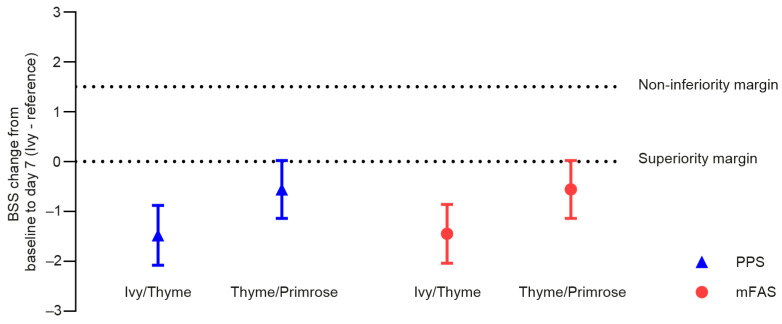
LS means treatment effect for BSS change from baseline to day 7 (Ivy—reference) and 95% confidence intervals for the per-protocol set (PPS) and the modified full analysis set (mFAS).

**Figure 3 pharmaceuticals-18-00754-f003:**
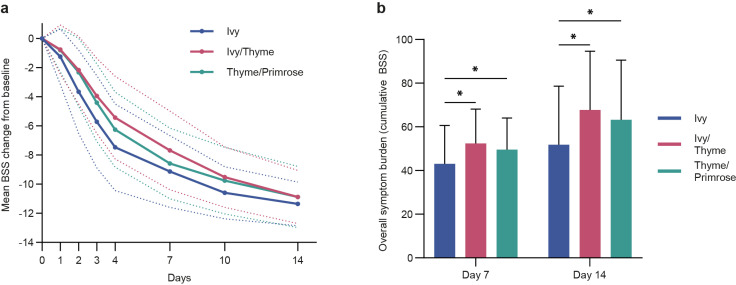
Analyses of Bronchitis Severity Score (BSS) changes between treatment groups over time. (**a**) Mean BSS change from baseline to all post-baseline visits with dashed lines indicating standard deviations (SD) for the modified full analysis set (mFAS). The differences between Ivy and Ivy/Thyme were statistically significant (*p* < 0.05) for all post-baseline visits. The differences between Ivy and Thyme/Primrose were statistically significant (*p* < 0.05) for all post-baseline visits except day 7 (*p* = 0.0598); (**b**) overall symptom burden at day 7 and day 14 displayed as cumulative BSS (mFAS). Results represent the mean + SD (* *p* < 0.05, compared to Ivy).

**Figure 4 pharmaceuticals-18-00754-f004:**
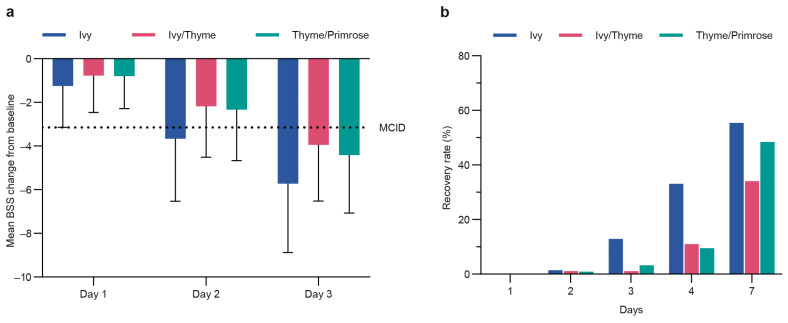
Bronchitis Severity Score (BSS) post hoc analyses. (**a**) BSS change from baseline to days 1, 2, and 3 with the minimal clinically important difference (MCID) threshold of 3.15 score points indicated with a dashed line for the modified full analysis set (mFAS); (**b**) recovery rates displayed as the proportion of patients with BSS less than 3 score points.

**Figure 5 pharmaceuticals-18-00754-f005:**
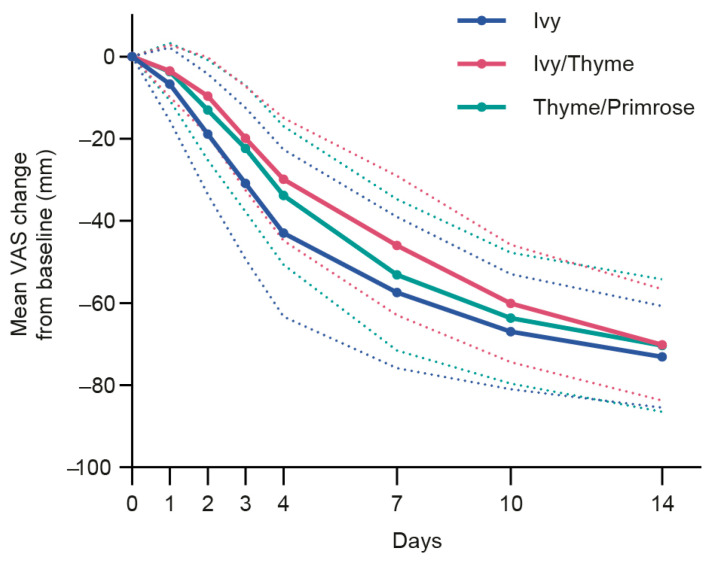
Patient-reported cough severity displayed as mean change of Visual Analog Scale (VAS) in mm from baseline to all post-baseline visits for the modified full analysis set (mFAS). Dashed lines indicate standard deviations. The differences between Ivy and each combination were significant for all post-baseline visits (*p* < 0.05).

**Figure 6 pharmaceuticals-18-00754-f006:**
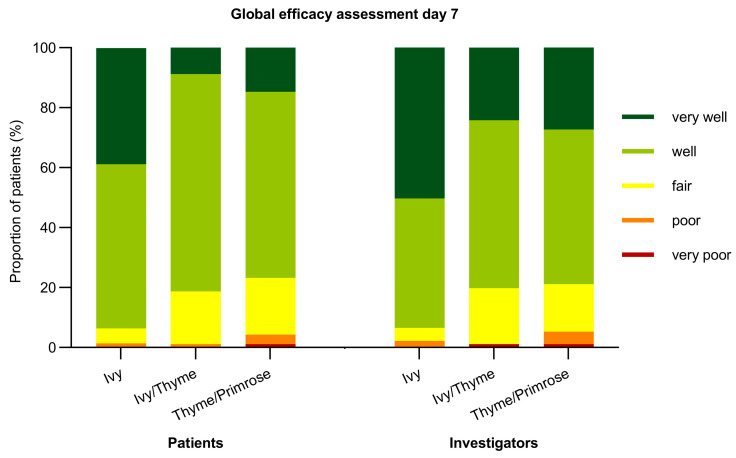
Assessment of global efficacy by patients (*considering all the ways this treatment has affected you since you started in the trial, how well are you doing?*) and investigators (*how do you rate this medication as a treatment for bronchitis?*) for the modified full analysis set (mFAS) on day 7.

**Table 1 pharmaceuticals-18-00754-t001:** Demographics and baseline characteristics (mFAS).

	Ivy (n = 139)	Ivy/Thyme (n = 91)	Thyme/Primrose (n = 95)
Age (years), mean ± SD	41.19 ± 13.56	40.92 ± 13.99	40.69 ± 14.87
Male, n (%)	57 (41.0)	44 (48.4)	40 (42.1)
BSS (points), mean ± SD	11.78 ± 1.39	11.71 ± 1.41	11.79 ± 1.42
Cough severity VAS (mm), mean ± SD	75.52 ± 11.67	75.84 ± 11.17	76.16 ± 11.10
Cough severity VCD categories, n (%)			
Distressing continuous coughing	8 (5.8)	1 (1.1)	6 (6.3)
Serious coughing	104 (74.8)	75 (82.4)	71 (74.7)
Frequent coughing	27 (19.4)	15 (16.5)	17 (17.9)
No cough	0 (0.0)	0 (0.0)	0 (0.0)

mFAS, modified Full Analysis Set; BSS, Bronchitis Severity Score; VAS, Visual Analog Scale; VCD, Verbal Category Descriptive.

**Table 2 pharmaceuticals-18-00754-t002:** Cough severity assessment by VCD categories (n (%), mFAS).

	Ivy (N = 139)	Ivy/thyme (N = 91)	Thyme/Primrose (N = 95)
Day 7			
Distressing continuous coughing	0 (0.0)	0 (0.0)	0 (0.0)
Serious coughing	4 (2.9)	2 (2.2)	2 (2.1)
Frequent coughing	16 (11.5)	25 (27.5)	12 (12.6)
Some short periods of cough	44 (31.7)	35 (38.5)	36 (37.9)
One short period of mild cough	43 (30.9)	24 (26.4)	37 (38.9)
No cough	32 (23.0)	5 (5.5)	8 (8.4)

mFAS, modified Full Analysis Set; VCD, Verbal Category Descriptive.

## Data Availability

The data presented in this study can be made available on a justified request from the corresponding author. The data are not publicly available due to privacy restrictions.

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
