# Peer review of "Efficacy and Safety of a Single Ivy Extract Versus Two Herbal Extract Combinations in Patients with Acute Bronchitis: A Multi-Center, Randomized, Open-Label Clinical Trial"

_pharmaceuticals, 2025, doi:10.3390/ph18050754_

Round 1

Reviewer 1 Report

Comments and Suggestions for Authors

The results of a study comparing the activity of a preparation containing a single plant extract with that of two combination preparations containing two different extracts each are presented. The results clearly indicated better therapeutic efficacy in the treatment of acute bronchitis of a preparation containing only a single ivy extract. The designed study, the way it was carried out, the presentation of the results of the study and its discussion do not raise any objections. However, several points should be clarified.

The plant drug products studied were Prospan® Cough Drops, whose name is mentioned only in Abstract and twice in section 4. Materials and Methods. In contrast, the manuscript used the term “ivy extract EA 575” (lines 25, 27, 61, 164, 167, 209 and 355), nowhere explaining why this was done. The authors should clearly define what they studied or explain why they introduced the new term “ivy extract EA 575” in place of “Prospan® Cough Drops.”

Keywords: there should also be the name “Prospan® Cough Drops” because this drug product was studied.

The Introduction section lacks information on the chemical profiles of the extracts studied. It is well known that the biological activity of plant extracts is the result of the synergistic action of all the active substances (secondary metabolites) contained in them. Therefore, when comparing the activity of three plant preparations (Prospan® Cough Drops, [Ivy], Bronchipret® Drops [Ivy/Thyme], and Bronchicum® Drops [Thyme/Primrose]), it is necessary to indicate (based on literature data) which groups of phytochemicals may be responsible for their biological activity.

In the manuscript, the term AUC is used. In clinical drug studies, AUC reflects the actual body exposure to drug after administration of a dose of the drug and is expressed in mg h/L. Since the manuscript is about clinical trials, it may dispense with the abbreviation “AUC” to avoid misunderstanding the results of these studies.

Line 156, please indicate how many as a percentage of the nine adverse events in the Thyme/Primrose group.

Author Response

Dear Reviewer,

We sincerely thank you for taking the time to carefully review our manuscript. We greatly appreciate the time and effort that you have dedicated to providing your valuable and detailed feedback on our manuscript, which has greatly helped to improve this paper. We have been able to incorporate changes to reflect most of the suggestions provided. Below is our point-by-point response to each comment:

Point 1:  The plant drug products studied were Prospan® Cough Drops, whose name is mentioned only in Abstract and twice in section 4. Materials and Methods. In contrast, the manuscript used the term “ivy extract EA 575” (lines 25, 27, 61, 164, 167, 209 and 355), nowhere explaining why this was done. The authors should clearly define what they studied or explain why they introduced the new term “ivy extract EA 575” in place of “Prospan® Cough Drops.” Keywords: there should also be the name “Prospan® Cough Drops” because this drug product was studied.

Response 1: We agree with you that the names of the investigated products should be explained more clearly. EA 575 is the active ingredient (a special ivy extract) contained in the product Prospan® Cough Drops. This information has been stated more clearly now in the abstract and methods section. We moreover thank you for suggesting to include the product name in the keywords, which we did along with the names of both comparator products. 

Point 2: The Introduction section lacks information on the chemical profiles of the extracts studied. It is well known that the biological activity of plant extracts is the result of the synergistic action of all the active substances (secondary metabolites) contained in them. Therefore, when comparing the activity of three plant preparations (Prospan® Cough Drops, [Ivy], Bronchipret® Drops [Ivy/Thyme], and Bronchicum® Drops [Thyme/Primrose]), it is necessary to indicate (based on literature data) which groups of phytochemicals may be responsible for their biological activity.

Response 2:  Thank you for pointing this out. We included these important information in the introduction (lines 57-61). Also please note our discussion regarding published pre-clinical experiments, which show that in addition to the variety and number of potentially active substances, these substances can still act antagonistically to each other in some cases (lines 228-237)   

Point 3: In the manuscript, the term AUC is used. In clinical drug studies, AUC reflects the actual body exposure to drug after administration of a dose of the drug and is expressed in mg h/L. Since the manuscript is about clinical trials, it may dispense with the abbreviation “AUC” to avoid misunderstanding the results of these studies.

Response 3: Thank you for helping us to avoid misunderstandings. We now replaced the term AUC in the manuscript, figure 3b, and corresponding figure legend with the description “cumulative BSS”.

Point 4: Line 156, please indicate how many as a percentage of the nine adverse events in the Thyme/Primrose group.

Response 4: Thank you very much for pointing out this missing information The results section has been updated accordingly.

Reviewer 2 Report

Comments and Suggestions for Authors

Pharmaceuticals (Manuscript ID: pharmaceuticals-3636837), Comments to the Authors:

Title: Efficacy and safety of a single ivy extract versus two herbal extract combinations in patients with acute bronchitis: a multi-center, randomized, open-label clinical trial

Comments

The submitted paper focused on the conduction of 3-arm, open-label, randomized clinical trial patients with acute bronchitis. The patients were assigned to groups receiving Ivy extract (Prospan® Cough Drops), Ivy/Thyme extract combination (Bronchipret® Drops), or Thyme/Primrose extract combination (Bronchicum® Drops) according to their respective labels. The primary endpoint was the assessment of non-inferiority, the second endpoint was the assessment of superiority of Ivy vs. each of the two comparators (Ivy/Thyme and Thyme/Primrose) regarding the change of Bronchitis Severity Score between baseline and day 7. 325 adult patients were considered for evaluation. Non-inferiority of Ivy extract was statistically significant against both comparators (both p < 0.0001). Superiority of Ivy extract was statistically significant against Ivy/Thyme extract (p < 0.0001) but missed statistical significance against Thyme/Primrose extract (p < 0.0607). The incidence of adverse events was low and comparable between the groups. The authors found that ivy extract EA 575 is non-inferior in acute bronchitis treatment compared to both comparators and superior to Ivy/Thyme.

I think the submitted manuscript can be accepted after the authors respond to the following comments:

  1. In the introduction, the authors should clarify why this head-to-head comparison of thyme-to-thyme combination therapies are critical. What are the gaps in the current guideline recommendations or clinical practice patterns that assume combination therapies are more effective.
  2. Potential biases are high in this clinical trial. Patient or investigator expectations are a source of bias, especially for subjective outcomes like the Bronchitis Severity Score (BSS) and Visual Analogue Scale (VAS). The authors should demonstrate how these biases were minimized in their study.
  3. Why did the authors use non-inferiority as the primary endpoint (followed by superiority). The selection was not clearly articulated in the manuscript.
  4. The authors used 3:2:2 randomization ratio is mentioned. However, they did provide the rationale for this unequal allocation.
  5. The conclusion does not give any indication on the beneficial uses for the current study on future therapeutic recommendation. How physicians can benefit from this study is important and should be clearly stated.
  6. The authors should expand their mentioned limitations to include the lack of placebo control and its implications for interpreting efficacy.

Author Response

Dear Reviewer,

We sincerely thank you for taking the time to carefully review our manuscript. We greatly appreciate the time and effort that you have dedicated to providing your valuable and detailed feedback on our manuscript, which has greatly helped to improve this paper. We have been able to incorporate changes to reflect most of the suggestions provided. Below is our point-by-point response to each comment:

Point 1:  In the introduction, the authors should clarify why this head-to-head comparison of thyme-to-thyme combination therapies are critical. What are the gaps in the current guideline recommendations or clinical practice patterns that assume combination therapies are more effective.

Response 1: A systematic literature review together with expert interviews recently suggested that the combination of ivy, primrose, and thyme extracts may contribute to multi-targeted treatment effects that have additive effects in cough treatment (Veldman et al. Pharmaceuticals (Basel). 2023;16(9):1206.).  Although this assumption may seem obvious, Veldman et al. also acknowledge that there is currently no evidence that the use of ivy, primrose, and thyme extracts in a mixture is preferable to the use of the three extracts separately. To make the rationale for our trial more clear, we have expanded on the current view of herbal combination therapies in the introduction (lines 45-50). Thank you for pointing this out and helping us to better articulate the current gap in scientific evidence that was directly assessed in our head-to-head trial.

Point 2: Potential biases are high in this clinical trial. Patient or investigator expectations are a source of bias, especially for subjective outcomes like the Bronchitis Severity Score (BSS) and Visual Analogue Scale (VAS). The authors should demonstrate how these biases were minimized in their study.

Response 2:  The BSS is an instrument which combines objective and subjective items, because the assessment is based on the investigator’s clinical evaluation in conjunction with the subjective feedback of the patient (Kardos et al. Pneumologie. 2014;68(8):542-546.). Since its introduction in 1996, the BSS has been successfully used in many clinical studies as a main outcome measure in patients suffering from acute bronchitis.

Nevertheless, we agree that patient or investigator expectations may be a source of bias given the open-label design of the trial. However, we believe that such bias did not influence the overall outcome of the trial due to the following reasons:

  1. As discussed in lines 253-257, the randomized allocation of patients to the study arms and the sufficiently large comparison groups minimized the likelihood of selective influence on the final results
  2. Our data is well in-line with the data from previous double-blind, placebo-controlled trials (see discussion in lines 208-224).

In order to further demonstrate how potential biases have been minimized, we added the information that investigators and site staff were trained on professional and impartial communication towards patients (lines 309-311).

Point 3: Why did the authors use non-inferiority as the primary endpoint (followed by superiority). The selection was not clearly articulated in the manuscript.

Response 3: The primary and secondary hypotheses of non-inferiority and superiority, respectively, were tested using the concept of hierarchically ordered hypotheses in an a priori defined order: e.g., superiority of Ivy extract vs. Ivy/Thyme extract was tested only if non-inferiority of Ivy extract vs. each of the two comparators could be shown (lines 324-327). Therefore, the trial design was set up to test for non-inferiority as a primary endpoint and then, if achieved, superiority as a secondary endpoint, which is generally considered as a valid trial design (U.S. Food & Drug Administration Non-Inferiority Clinical Trials to Establish Effectiveness, January 2025 Update, https://www.fda.gov/media/78504/). We aimed to make the rationale for our study design more clear by revising the final section of our introduction (lines 69-76). 

Point 4: The authors used 3:2:2 randomization ratio is mentioned. However, they did provide the rationale for this unequal allocation.

Response 4: Thank you for pointing out that additional information is required regarding the allocation ratio. Such unequal ratios in favor of the comparator-group verus both reference groups are common in statistical experimental design. Even with unbalanced sample sizes, the significance statements (p-values) generated by significance tests are valid. This is already explained in the methods section (lines 349-353). Furthermore, we now provide additional information in the results section to underscore the rationale for comparing the monotherapy as the reference with the two combination products, so that a total of 140 Ivy patients were compared with overall 188 non-Ivy patients (lines 79-82).

Point 5: The conclusion does not give any indication on the beneficial uses for the current study on future therapeutic recommendation. How physicians can benefit from this study is important and should be clearly stated.

Response 5: We would like to emphasise that our study cannot replace therapeutic recommendations from clinical guidelines. As all three investigated preparations are explicitly recommended as possible treatment options in clinical guidelines (see lines 66-67), the current trial is not designed to replace such recommendations.

However, in practice, it is often assumed that combinations of extracts intuitively offer efficacy advantages over single extracts because of the expectation of an additive effect (see answer 1 and lines 45-50). Our data clearly show that mono extracts can be at least as effective or even  superior in some cases. We believe that this finding is of fundamental interest to the scientific community and may moreover be taken into account in therapeutic considerations. Together, we believe that the direct implications of our data for both the scientific community and physicians are sufficently discussed in the conclusions section (lines 372-379).

Point 6: The authors should expand their mentioned limitations to include the lack of placebo control and its implications for interpreting efficacy.

Response 6: The general efficacy of all three investigated extracts/extract combinations has  already been shown in placebo-controlled clinical trials (see lines 65-66). Therefore, the aim of the study was not showing efficacy over placebo but comparing a mono ivy extract with herbal combination products. We now provided additional information to the discussion (lines 238-242).

Round 2

Reviewer 2 Report

Comments and Suggestions for Authors

Pharmaceuticals (Manuscript ID: pharmaceuticals-3636837), Comments to the Authors:

Title: Efficacy and safety of a single ivy extract versus two herbal extract combinations in patients with acute bronchitis: a multi-center, randomized, open-label clinical trial

Comments

After reading the authors response to my comments, I think the revised manuscript can be accepted for publication.